



# Measurement report: Atmospheric mercury in a coastal city of
# Southeast China: inter-annual variations and influencing factors
Jiayan Shi[1,2,4], Yuping Chen[1,2,3], Lingling Xu[1,2,*], Youwei Hong[1,2], Mengren Li[1,2], Xiaolong Fan[1,2], Liqian
Yin[1,2],Yanting Chen[1,2], Chen Yang[1,2,3], Gaojie Chen[1,2,3], Taotao Liu[1,2,3], Xiaoting Ji[1,2,3], Jinsheng Chen[1,2,*]
[1] Center for Excellence in Regional Atmospheric Environment, Institute of Urban Environment, Chinese
Academy of Sciences, Xiamen 361021, China
[2] Key Lab of Urban Environment and Health, Institute of Urban Environment, Chinese Academy of
Sciences, Xiamen 361021, China
[3] University of Chinese Academy Sciences, Beijing 100049, China
[4] College of Resources and Environment, Fujian Agriculture and Forestry University, Fuzhou 350002,
China
**Correspondence:** Lingling Xu (linglingxu@iue.ac.cn) and Jinsheng Chen (jschen@iue.ac.cn)





**Abstract.** Long-term monitoring of atmospheric mercury is an important part of the effective evaluation of the Minamata Convention on Mercury. Gaseous elemental mercury (GEM) along with conventional air pollutants and meteorological parameters were simultaneously observed in Xiamen city, Southeast China in January and July over the period 2012 – 2020. GEM concentrations in January slightly increased from 2012 (3.50 ng m$^{-3}$) to 2015 (4.47 ng m$^{-3}$) and then decreased to 2020 (3.93 ng m$^{-3}$), while GEM in July maintained stable from 2012 to 2017, and decreased significantly in 2020. The temporal variation of GEM was characterized by higher concentrations in winter than in summer and in nighttime than in daytime. Bivariate polar plots and HYSPLIT model were used to identify the source regions of GEM on a local and regional scale. The results indicate that GEM concentrations of air masses from all directions in January 2015 were higher than those of other years. Generalized Additive Models (GAMs), a regression analysis method, were established and applied to investigate the influencing factors on the inter-annual trend of GEM. The factors, anthropogenic emissions, meteorological conditions and transmission explained 26.9 ± 11.4%, 45.7 ± 7.2% and 27.4 ± 6.8% to the variation of GEM concentrations, respectively. Among them, the interpretation rate of anthropogenic emissions has generally decreased since 2012, whereas meteorology explained the largest proportion of GEM concentrations in almost all study periods. Our results suggest that meteorology was the more important factor, which driven the inter-annual trend of GEM in the study region.

**Keywords:** Inter-annual trend; Gaseous elemental mercury; GAMs; HYSPLIT model; Meteorology factors.





## 1 Introduction


Atmospheric mercury, because of its neurotoxicity, long persistence and high bioaccumulation, is
defined as a global pollutant that poses a threat to the health of the global population. Especially the
organic form of mercury, like methylmercury (MeHg), is associated with neurocognitive deficits in
human fetuses and cardiovascular effects in adults (Axelrad et al., 2007; Roman et al., 2011). In order to
protect human health and the environment from adverse effects of mercury, a legally-binding
international treaty Minamata Convention on Mercury was adopted in October 2013 and entered into
force in August 2017 (UN Environment, 2017). The Minamata Convention requires parties to reduce
mercury emissions and to assess the effectiveness of mitigation measures. China is one of the world's
largest mercury emitters as well as the signatory to the Minamata Convention. The annual atmospheric
mercury emissions in China were about 565 tons in 2015, accounting for about a quarter of global
anthropogenic mercury emissions (AMAP/UNEP, 2018). The predominant anthropogenic Hg emission
sources in China were industrial coal combustion, coal-fired power plants, nonferrous metal smelting,
and cement production (Zhang et al., 2015). The characteristics of Hg emissions such as large emission
amount and industrial production-dominated emissions made that China has great potential to reduce
mercury emissions through the implementation of the Minamata Convention on Mercury. A mercury
emissions inventory of China for 1978 – 2014 has reported that anthropogenic mercury emissions peaked
in 2011 and then showed a downward trend (Wu et al., 2016). In addition, China had reduced
anthropogenic mercury emissions by 127 tons from 2013 to 2017 (Liu et al., 2019b). Among them,
mercury emissions from the coal-fired power plants fell from 105 tons in 2007 to 73 tons in 2015 (Zhang
et al., 2015; Liu et al., 2018). The main reason for this downward trend was that China has introduced a
series of air cleaning measures including upgrading precipitator devices, newly built desulfurization and
denitrification devices and ultra-low emission renovations since 2013, which led to synergistic removal
of mercury (Liu et al., 2019b). Strict restrictions of mercury on the mining, production, utilization, import
and export have also been imposed in China since 2017 (https://www.mee.gov.cn/, last access: 17 May

2022).



Atmospheric mercury is usually classified into three categories according to the determination
technique: gaseous elemental mercury (GEM), gaseous oxidized mercury (GOM) and particulate bound
mercury (PBM). Because of its stability and volatility, GEM is the dominant form of atmospheric
mercury, accounting for up to 95%. GEM has an atmospheric residence time of $0.5 - 2$ years and can
spread globally before being deposited to earth's surfaces (Schroeder and Munthe, 1998; Zhang et al.,
2013; Yuan et al., 2021). On the other hand, GOM and PBM have relatively high reactivity and dry/wet
deposition rates and are therefore easier to be removed from the atmosphere. The sum of GEM and GOM
is referred as total gaseous mercury (TGM), but GOM contributed generally less than 5% of TGM
(Schroeder and Munthe, 1998; Xu et al., 2020). Thus, the GEM concentrations were usually
approximated to TGM concentrations. Field observations are vital to understand the long-term variation
of Hg levels in the atmosphere. There are some global and regional mercury monitoring networks around
the world which provided a long-term monitoring result. For example, a steep decline trend of GEM
concentrations ($0.05$ ng m$^{-3}$ yr$^{-1}$) was observed at Mace Head, Ireland, from 2013 to 2018 (Custodio et
al., 2020), while GEM data from Cape Point, South Africa, showed a slight increase from 2007 to 2017
(Slemr et al., 2020). China has also carried out a series of observational studies in the past two decades.
These observation studies on GEM concentrations mostly focused on $1 - 2$ years, while few continuous
GEM observation records over multiple years in China were published (Fu et al., 2015). A three-year
measurement at Chongming Island, East China, showed that the annual GEM concentrations
significantly decreased from $2.68$ ng m$^{-3}$ in 2014 to $1.60$ ng m$^{-3}$ in 2016, at a rate of $-0.60 \pm 0.08$ ng m$^{-3}$
$^{3}$yr$^{-1}$ (Tang et al., 2018). Whereas, a multi-year observation of GEM in Guiyang, Southwest China,
showed an increasing trend from $8.40$ ng m$^{-3}$ in 2002 to $10.2$ ng m$^{-3}$ in 2010, and the increase mainly
occurred during the cold season (Fu et al., 2015).
The variation of GEM concentrations is influenced by a variety of factors such as anthropogenic
emissions, meteorological conditions, as well as intra- and inter-regional transport (Tang et al., 2018; Liu
et al., 2019a; Zhang et al., 2021). In previous studies, the impact of anthropogenic emissions changes
was often quantified by compiling emission inventories (Zhang et al., 2015; Wu et al., 2016; Liu et al.,
2019b; Cai et al., 2020). The trajectory-based analysis method was frequently applied to analyze the
impact of regional transport (Tang et al., 2018; Wang et al., 2021). However, the results like the impacts



of anthropogenic emissions and regional transport derived from above different method systems could
not be comparable. Generalized additive models (GAMs) have been introduced into influencing factor
identification in recent years, which are data-driven and able to incorporate non-linear relationships of
air pollution with numerical and categorical variables (Wood and Augustin, 2002). The impact of local
anthropogenic emissions, regional transport, and meteorological factors on GEM concentrations in
Nanjing, East China was quantified by using the GAMs (Zhang et al., 2021). GAMs were also used to
explain the decline of GEM concentrations in Beijing and the result showed that reduction of
anthropogenic mercury emissions, variation in meteorological conditions, and change in globe
background level explained 51.5%, 47.1% and 1.4% of the decrease of GEM concentrations, respectively
(Wu et al., 2020). It can be seen that GAMs are a promising tool to explore the effect of factors like
anthropogenic emissions and natural perturbations on GEM concentrations.

China has adopted aggressive atmospheric control measures in the last decade. Long-term GEM

observation was very necessary to investigate the variation of GEM levels and its influencing factors. In
this study, GEM, conventional pollutants and meteorological parameters were simultaneously observed
in Xiamen, a coastal city in Southeast China, in January (represents winter) and July (represents summer)
over the period 2012 – 2020. The main objectives of this study are: (1) to characterize the inter-annual,
seasonal and diurnal variations of GEM in a coastal city of Southeast China, (2) to identify the source
regions of GEM on a local and regional scale and their influence on annual concentrations of GEM, (3)
to investigate the influencing factors including anthropogenic emissions, regional transport and
meteorology on the inter-annual variation of GEM concentrations.
**2 Method**
**2.1 Site description**

The study site (Xiamen, 118°04'13''E, 24°36'52''N) is located in the Institute of Urban Environment,

Chinese Academy of Sciences in Jimei District of Xiamen City, Fujian Province, China (Fig. 1a). The
site was characterized by a typical subtropical monsoon climate, with the prevailing ocean monsoon in
summer and northerly or northeasterly winds from the inland of China in winter. Industrial point sources
were mainly distributed to the northeast and the southwest of the study site (Fig. 1b). The instruments



were placed on the roof of a building (~80 m above the ground). The outdoor air inlet of the sampling
unit was located at 2 m above the rooftop of the building.

**Figure 1. (a) The location of study site in Xiamen City, Fujian Province, China and the regional distribution**


**map of anthropogenic Hg emissions in China in 2014 (Wu et al., 2016). Note that the red dots represent the**






**amount of mercury emitted by each province. (b) The distribution of local industrial point sources in Xiamen.**
**Note that the colors of the dots represent different industrial categories, and the size represents a company's**
**output or coal consumption (ton yr⁻¹).**

### 2.2 Atmospheric mercury measurements

Atmospheric mercury was measured by the Tekran 2537B/1130/1135 system (Tekran Inc., Toronto,
Canada). Continuous 5 min of GEM concentrations were measured by a Tekran 2537B Hg vapor analyzer,
with a detection limit of 0.06 ng m$^{-3}$ at a sampling flow rate of 1 L min$^{-1}$ (Tekran, 2001; Xu et al., 2015).
The principle and the routine maintenance of the equipment in detail have been described in the previous
study (Xu et al., 2015). The Tekran 2537B analyzer was calibrated automatically every 25h using the
internal Hg permeation source inside the instrument, and the accuracy of this permeation source was
calibrated every 12 months with manual injection of Hg by a syringe from an external Hg source (Module
2505). This study was based on the GEM observation data of January and July (representative of winter
and summer, respectively) in 2012, 2013, 2015, 2017 and 2020. Gap years in GEM data were generally
due to abnormal operation of the instrument or mismatch observation periods. In order to match other
parameters, the time resolution of GEM concentrations was converted from the original 5 min to 2 h. The
proportion of GOM in Xiamen was less than 5% of the TGM during the study period. Thus, GEM
concentrations in this study were directly compared to TGM in the below analysis.

### 2.3 Meteorological parameters and criteria air pollutants

In this study, the conventional pollutants (including SO$_2$, NO$_2$, O$_3$, CO, PM$_{10}$, PM$_{2.5}$) and
meteorological parameters (including wind speed (WS), wind direction (WD), relative humidity (RH),
air temperature ($T$) and surface air pressure (SP)) were obtained from Xiamen air quality monitoring
station. Note that above pollutants concentrations and meteorological data were averaged into 2 h time
intervals. Other meteorological parameters: boundary layer height (BLH), downward UV radiation at the
surface (UVB) and low cloud cover (LCC) were obtained from the European Centre for Medium-Range
Weather Forecasts (ECMWF) reanalysis (https://www.ecmwf.int, last access: last access: 23 March

2022).



**2.4 Potential source regions identification**
Hybrid Single-Particle Lagrangian Integrated Trajectory (HYSPLIT) model and Global Data
Assimilation System data (https://www.arl.noaa.gov/, last access:5 May 2022) were applied to calculate
the 72h backward air mass trajectories. The interval of backward trajectories was 2 h and the arrival
height was set as 500 m above the ground level. Total spatial variance (TSV) method was chosen to
calculate clusters in the HYSPLIT calculation (Draxler, 1999). The clusters of air mass trajectories were
further categorized into five potential source regions in January and four source regions in July according
to their travel paths. For example, air parcels which originated from Fujian province over the last 3 d
were considered to be local air masses named Local (the details of the source region definition are
presented in Support Information). The observed GEM concentrations were assigned to the trajectories.
The mean GEM concentration related to each potential source region ($C_i$) was calculated by Equation
(1). The weighted GEM concentration contribution ($W_{Regions}$ ) of each source region to the observed
average GEM was calculated by Equation (2):

$$C_i = \frac{\sum_{l=1}^{M_i} C_l}{M_i}, \tag{1}$$

$$W_{Regions} = \frac{\sum_{l=1}^{M_i} C_l}{\sum_{i=1}^{n} C_i \times M_i}, \tag{2}$$

Where $i$ is the index of the source regions, $l$ is the index of the trajectory, $M_i$ is the total number of
trajectories originated from the $i$ source regions, $C_l$ is GEM concentration observed upon arrival of
trajectory $l$, $n$ is the number of all source regions over each time period.
**2.5 Model development**
Generalized Additive Models, a regression analysis method, have been used to establish the
relationship between GEM and various variables, and to investigate the influencing factors on the inter-
annual trend of GEM concentrations (Gong et al., 2017). The models were run by the following steps:
model establishment, parameter selection and model quality control.
**Model Establishment:** GAMs were performed using R version 4.1.2 with the "mgcv" package. The
equation can be described as follows:

$$g(\mu) = f_1(x_1) + f_2(x_2) + \cdots + f_k(x_k) + \varepsilon, \tag{3}$$

Where $x_j$ (j = 1, 2, 3, ..., k) are different meteorological predictors and $f_j$ is a smooth function of the predictors; $\varepsilon$ is the residual; $\mu$ is the expected value of the response variable; and $g$ is the link function specifies the relationship between the non-linear formulation and the expected value. We used the "identity link" function with a Gaussian distribution because the relationship between GEM and the variables conformed to a Gaussian distribution and the estimation of GAMs was considered unbiased. In order to ensure the balance between under-fitting and overfitting of observation data, we used a penalized cubic regression as a smooth function. In the running process of the model, the concentration contribution of the smoothed independent variable to the dependent variable was output and converted into contribution ratio, which is helpful to determine the degree of each variable driving the prediction.

**Parameter Selection:** In this study, 16 variables accompanied with GEM concentrations were used for GAMs establishment. These variables were divided into four categories: anthropogenic emissions ($SO_2$, $NO_2$, $O_3$, CO, $PM_{2.5}$, $PM_{10}$), surface meteorology ($T$, RH, WS, WD, SP), high-altitude meteorology (BLH, UVB and LCC) and air transmission (24h-Latitude and 24h-Longitude) (Table S1). In order to eliminate the significant colinearity variables, the colinearity diagnosis method was adopted to make judgment according to the variance inflation factor (VIF). The performance of GAMs was judged according to Akaike Information Criterion (AIC) and $R^2$ values. Specifically, as the parameters were successively added into the model, the AIC decreased and $R^2$ increased. Based on this method, 5 variables including CO, RH, SP, 24h-Latitude and 24h-Longitude were eventually selected into the model (Table S2). Given that the main anthropogenic sources of GEM in China are combustion (Liu et al., 2019b; Wu et al., 2020), CO was used to represent the anthropogenic Hg emissions factor. RH and SP were classified as the meteorological factor, and 24h-Latitude and 24h-Longitude represented the transmission factor (Wu et al., 2020). Given that five selected variables passed the colinearity test, the three factors in the model were considered to be independent of each other.

**Model Quality Control:** The accuracy of GAMs simulation was assessed using a 10-fold cross-validation test. The principle of the test is dividing the whole dataset into ten subsets randomly, and in each round of cross-validation, nine subsets are used to fit the model and the remaining one is predicted. This process is repeated 10 times to ensure that every subset is tested. The 10-fold cross-validation results





showed a good coincidence between the GAMs and cross-validated results ($R^2$ = 0.97, Fig. S1),
demonstrating the reliability of the established model. In order to test the underlying assumptions of
homogeneity, normality and independence of GAMs to ensure the validity and accuracy of the model,
we used the following methods (Fig. S2): (1) Quantile-quantile (QQ) plots (Sample quantiles against
theoretical quantiles), (2) scatterplots (residuals against linear predictor), (3) histograms of the residuals
and (4) scatter plots (response against fitted value). The QQ-plot showed that the GAMs produced good
results around the average concentrations and the residuals showed a normal distribution. The fitted GEM
and observed GEM were also compared to further valid the accuracy of the established model.
**3 Results and discussion**
**3.1 Temporal variations of GEM concentrations**
**3.1.1 Inter-annual variation**
Monthly concentrations of GEM ranged from 3.50 to 4.47 ng m$^{-3}$ in January and 1.56 to 2.65 ng m$^{-3}$
in July during the whole study period, with mean values of 4.04 ± 1.01 ng m$^{-3}$ and 2.29 ± 0.83 ng m$^{-3}$,
respectively. The GEM concentrations in Xiamen were several times higher than the Hemisphere
background concentrations (about 1.5 – 1.7 ng m$^{-3}$) (Lindberg et al., 2007; Sprovieri et al., 2010).
Comparisons of TGM/GEM concentrations in Xiamen with other urban and rural areas in East Asia over
the last decade are shown in Table 1. The mean concentrations of GEM mostly fell in a range of 2 – 5 ng
m$^{-3}$ in East Asia (except for some background sites). The GEM in Xiamen was slightly higher than those
measured at rural and background monitoring sites such as Tibetan Plateau region, Mt. Changbai, and
Mt. Ailaoshan (Zhang et al., 2016a; Yin et al., 2018; Liu et al., 2019a), while lower than those reported
from inland urban sites like Lanzhou, Nanjing, and Shanghai (Zhu et al., 2012; Duan et al., 2017; Yin et
al., 2020).



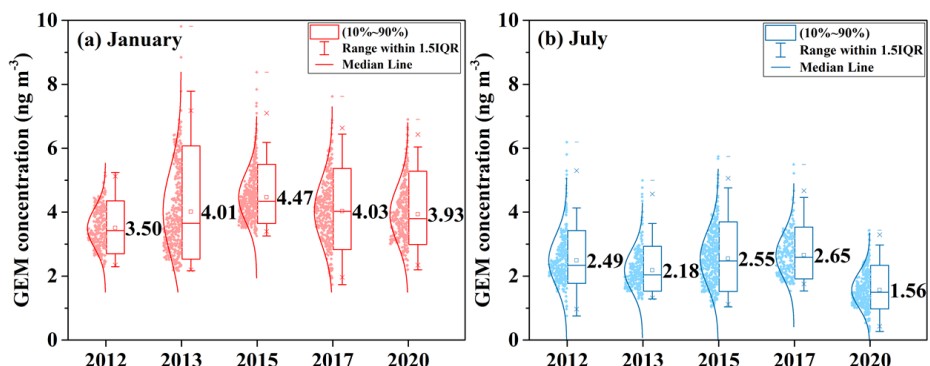


**Figure 2. Statistical summaries of gaseous elemental mercury (GEM) concentrations in Xiamen in (a) January**
**and (b) July of the study years.**






**Table 1. Comparisons of TGM/GEM concentrations in Xiamen with other urban and rural areas in East Asia over the period 2010 – 2020.**

| Locations | Classification | Time | TGM/GEM (ng m⁻³) | Winter | Summer | References |
|---|---|---|---|---|---|---|
| Xiamen, CN | Suburban (Coast) | 03/2012–02/2013 | 3.50±1.21 | / | / | (Xu et al., 2015) |
| Mt. Changbai, CN | Rural | 07/2013–07/2014 | 1.68±0.47 | / | / | (Liu et al., 2019a) |
| Haidian, CN | Urban | 02/2018–03/2018 | 2.77±0.91 | / | / | (Wang et al., 2021) |
| Shanghai, CN | Suburban | 06/2014–12/2014 | 4.19±9.13 | 5.5±6.6 | 3.8–5.3 | (Duan et al., 2017) |
| Shanghai, CN | Suburban | 12/2015–02/2016 | 2.77±1.36 | 2.88 | 2.87 | (Qin et al., 2019) |
| Hefei, CN | Suburban | 07/2013–06/2014 | 4.07±1.91 | 4.05 ± 1.81 | 4.08±1.99 | (Hong et al., 2016a) |
| Ningbo, CN | Coastal | 04/2011–04/2013 | 3.30±1.40 | 3.70 | 2.60 | (Yu et al., 2015) |
| Ningbo, CN | Urban (Coastal) | 07/2013–01/2014 | 3.26±1.63 | / | / | (Hong et al., 2016b) |
| Ningbo, CN | Urban (Coastal) | 12/2016–11/2017 | 2.44±0.95 | 2.62±1.05 | 2.26±0.78 | (Yi et al., 2020) |
| Nanjing, CN | Urban | 01/2011–12/2011 | 7.94±6.99 | 5.5±2.5 | 9.9±8.2 | (Zhu et al., 2012) |
| Mt. Ailaoshan, CN | Rural | 05/2011–05/2012 | 2.09±0.63 | 2.04±0.58 | 2.20±0.60 | (Zhang et al., 2016a) |
| Taoyuan city, Taiwan, CN | Suburban | 10/2017–09/2018 | 2.61±6.47 | 2.19±0.75 | 2.27±1.67 | (Sheu et al., 2019) |
| Taichung city, Taiwan, CN | Rural | 10/2014–09/2015 | 1.19 | / | / | (Fang et al., 2017) |
| Yongheung, Korea | Island | 01/2013–08/2014 | 2.80±1.10 | 3.50–3.70 | 2.30±0.90 | (Lee et al., 2016) |
| Gyodong Island, Korea | Rural | 08/2015–09/2017 | 2.70±2.60 | 2.8±2.9 | 1.70±1.0 | (Lee et al., 2019) |
| Fukuoka, Japan | Urban | 06/2012–05/2013 | 2.33±0.49 | 2.31±0.44 | 2.37±0.58 | (Marumoto et al., 2015) |
| Nam Co, Tibetan Plateau, CN | Plateau | 01/2012–10/2014 | 1.33±0.24 | 1.14±0.18 | 1.50±0.20 | (Yin et al., 2018) |
| Changdao Island, CN | Rural (Coastal) | 10/2013–07/2015 | 2.52±0.82 | 2.87±1.16 | 2.25±0.51 | (Wang et al., 2020) |
| Lanzhou, CN | Urban | 10/2016–10/2017 | 4.48±2.32 | 5.06±2.45 | 4.45±2.10 | (Yin et al., 2020) |





The inter-annual variability of GEM concentrations in January and July are shown in Fig. 2. The GEM
concentrations in Xiamen showed no distinct trends over the period 2012 – 2020. Specifically, GEM
concentrations in January displayed a slight upward trend from 2012 (3.50 ng m$^{-3}$) to 2015 (4.47 ng m$^{-3}$)
and a decreasing trend from 2015 to 2020 (3.93 ng m$^{-3}$). Whereas GEM concentrations in July were stable
from 2012 to 2017, and decreased significantly in 2020. Hg emission inventories showed that the total
anthropogenic Hg emissions in China were mitigated during the last decade and the inflection point was
most likely to occur between 2010 and 2015 (Zhang et al., 2015; Wu et al., 2016; Liu et al., 2018; Liu et
al., 2019b). Recent studies have indicated either a stable or a slight decreasing trend for GEM or TGM
concentrations in Chinese cities after 2013 when China has applied the aggressive measures to control
air pollution (Qin et al., 2020; Wu et al., 2020; Yin et al., 2020). For instance, it was reported that GEM
concentrations at Chongming Island in East China significantly decreased from 2014 to 2016, and the
inflection point occurred before 2014 (Tang et al., 2018). Note, those measurements mostly lasted for 2
– 4 years. So far, the observations of GEM concentrations over a long time period were scarce. Our result
suggests the influencing factors on the variation trend of GEM in East China would be complex over the
last decade.
Coal combustion is one of the leading mercury sources in China (Wu et al., 2006). Table S3 summaries
the consumption of coal and the statistics of annual SO$_2$ and NO$x$ emissions in Fujian Province over the
period 2012 – 2020. The coal consumption in Fujian Province showed a small fluctuation among years.
Whereas, there was a decreasing trend for SO$_2$ and NO$x$ emissions from 2013 to 2017 with the air
pollution control measures implemented in China. Previous studies have found that mercury can be
synergistically removed in the process of desulfurization and de-nitration (Zhang et al., 2016b; Liu et al.,
2019b). The installation of selective catalytic reduction to control nitrogen oxide emissions is often
accompanied by the oxidation of GEM to GOM, and the combined application of selective catalytic
reduction and flue gas desulfurization could further reduce TGM emissions to the atmosphere (Rallo et
al., 2012). Nonetheless, the inconsistent inter-annual trend in GEM concentrations and SO$_2$/NO$x$
emissions indicates that additional factors, like GEM emission sources other than coal combustion and/or
meteorological changes drove the inter-annual variation trend of GEM in the study region.



### 3.1.2 Seasonal and diurnal patterns

The GEM concentrations in Xiamen were approximately 1.41 – 2.52 times higher in winter than in summer ($P < 0.001$, one-way ANOVA) over the study years. The similar seasonal variation was widely observed in cities including Shanghai, Ningbo, Lanzhou, and Yongheung, as well as Gyodong Island, and Changdao Island (Yu et al., 2015; Hong et al., 2016b; Lee et al., 2016; Duan et al., 2017; Lee et al., 2019; Wang et al., 2020; Yi et al., 2020; Yin et al., 2020). However, a reverse seasonal variation with higher GEM in summer than in winter was observed in Nanjing, Chongming Island, Mt. Ailaoshan and Tibetan Plateau region (Zhu et al., 2012; Zhang et al., 2016a; Tang et al., 2018; Yin et al., 2018). There were many factors responsible for the seasonal variation of GEM in Xiamen. In terms of Hg emission sources, local industrial emissions were relatively stable over the course of a year. The key factor with seasonal changes is the increased usage of coal for heating which mainly occurred in northern China in cold seasons. Although there was no coal consumption for heating in southern China, GEM is well mixed due to its prolonged lifetime (0.5 – 1 year) (Qiu et al., 2021). Monsoonal winds can change the source–receptor relationships at observation sites, and thus affect the seasonal variation of GEM concentrations (Fu et al., 2015; Liu et al., 2019a). Winter winds in Xiamen mainly originated from north directions which passed through numerous intensive anthropogenic GEM emissions areas (Fig. 1a), while summer winds mainly originated from ocean with less GEM point sources. Another important factor is that the mixing heights were reduced due to stable inversion layer in winter. As a result, GEM diffused slowly and accumulated easily in the surface layer. In addition, for the sites in the Northern Hemisphere, the greater removal of GEM by wet and dry deposition could also lead to lower GEM concentrations in warmer seasons (Fu et al., 2008; Tang et al., 2018).

The diurnal variations of bihourly GEM concentrations were consistent among years (Fig. 3). The GEM concentrations generally displayed a downward trend during the day and an accumulation trend over time at night, with a peak at 8:00 – 10:00 am and a valley at 14:00 – 16:00 pm. Among them, GEM concentrations in January 2015 were relatively high on the whole, and the diurnal change of GEM concentrations was gentle, which might be related to the enhanced transport of continental pollution from northeast Asia due to the extreme 2015 – 2016 El Niño event (Fu et al., 2012; Nguyen et al., 2022). The diurnal pattern of GEM concentrations in Xiamen is consistent with other urban sites like Guiyang, Hefei



and Guangzhou (Feng et al., 2004; Chen et al., 2013; Fu et al., 2015). Previous studies often attributed
diurnal variations of GEM to the effect of various anthropogenic emissions, photochemical oxidation
and the diurnal variation of BLH (Hong et al., 2016b; Duan et al., 2017). The diurnal pattern of GEM in
Xiamen was similar to those of $SO_2$, $NO_2$ and CO (Fig. S3a – c), reflecting the combined effects of
common anthropogenic emissions and the diel fluctuation in meteorology such as the BLH. The
decreasing trend of GEM from early morning to afternoon was due to the intensified turbulent mixing in
the boundary after sunrise while the nighttime had the opposite condition (Fig. S3e). In addition, the
GEM concentrations decreased during the daytime with the increase of $O_3$ (Fig. S3d). Thus, we could
expect that the photo-oxidation of GEM to GOM partly reduced the GEM concentrations after morning.

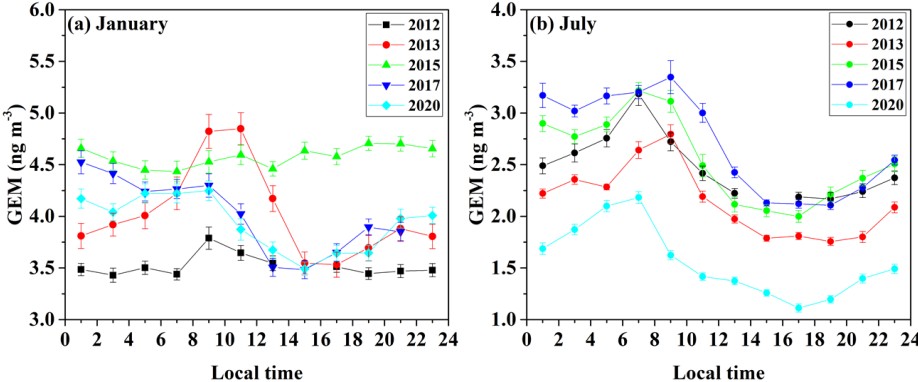


**Figure 3. The diurnal trend of GEM concentrations in (a) January and (b) July over the study years. Note**
**that the value of error bars has been reduced tenfold.**
**3.2 Potential source regions of GEM**
**3.2.1 Local emissions**
According to the industrial Hg emissions inventory in China, the main Hg emissions industries
included industrial coal combustion, coal-fired power plants, non-ferrous metal smelting, cement
production, and waste incineration (Zhang et al., 2015; Wu et al., 2016). The relationships between the
spatial distribution of industrial sources and bivariate polar plots of GEM concentrations would shed
light on the influence of local anthropogenic point sources on GEM concentrations. As shown in Fig.4a,
4d, the elevated GEM concentrations in the bivariate polar plots of individual seasons were highly
concentrated, suggesting that the spatial distribution of main Hg point sources was similar among years.





However, the polar plot results of GEM concentrations were distinctly different between seasons likely
due to the shift in the wind. In January, the elevated GEM concentrations were associated with west wind
with a low WS of 0.5 – 2.0 m s$^{-1}$ (Fig. 4a), which indicates nearby emission sources. They were likely
the industrial coal combustion, and non-ferrous metal smelting upwind from the west of the monitoring
site (Fig. 1b). The conventional pollutants are the good indicators of primarily anthropogenic sources. In
Xiamen city, the coal-fired power plants contributed 62% of local SO$_2$ emissions and 57% of CO
emissions. A close correlation of GEM with SO$_2$ and CO in westerly WD with low speeds (Fig. 4b, 4c)
further supported above conclusion that the contributions of local industrial sources to GEM.
In July, the elevated GEM concentrations occurred when winds came from the southwest with the WS
about 2.5 – 3 m s$^{-1}$ (Fig. 4d). As shown in Fig. 1b, there are many industrial clusters in the southwest
direction of the observation site including coal-fired power plants, industrial coal combustion and
nonferrous metal smelting. Accordingly, we suspected that the local industrial clusters upwind of the
southwest to the study site caused an evident increase in GEM concentrations. High WS of the southwest
wind likely weakened the correlation between GEM and SO$_2$ (Fig. 4e) while GEM and CO remained a
good correlation in the southwest wind due to their stable chemical properties (Fig. 4f). Another elevated
GEM concentration condition occurred when wind came from the east with a lower WS of 0 – 2 m s$^{-1}$
(Fig. 4d). Such low WS suggests a stagnant meteorological condition which was unfavorable for GEM
dispersion. In addition, GEM and SO$_2$ showed a good correlation in the case of the east wind with WS
of 1 – 2 m s$^{-1}$. Hence, we could speculate that the upwind point sources, like industrial coal combustion
and nonferrous metal smelting, as well as the adverse atmospheric diffusion conditions contributed to the
increasing GEM concentrations. As shown in Table S4, GEM in Xiamen was overall positively correlated
with SO$_2$, NO$_2$, CO and PM$_{2.5}$, but the correlation coefficient fluctuated remarkably among years. In
addition, the inter-annual trend of GEM concentrations was not coincided with those of SO$_2$ or NO$_x$
emissions as mentioned above. Thus, although local Hg point sources contributed to the elevated GEM
concentrations in individual seasons, they might be not the dominant factor on the inter-annual trend of
GEM concentrations in the study region.



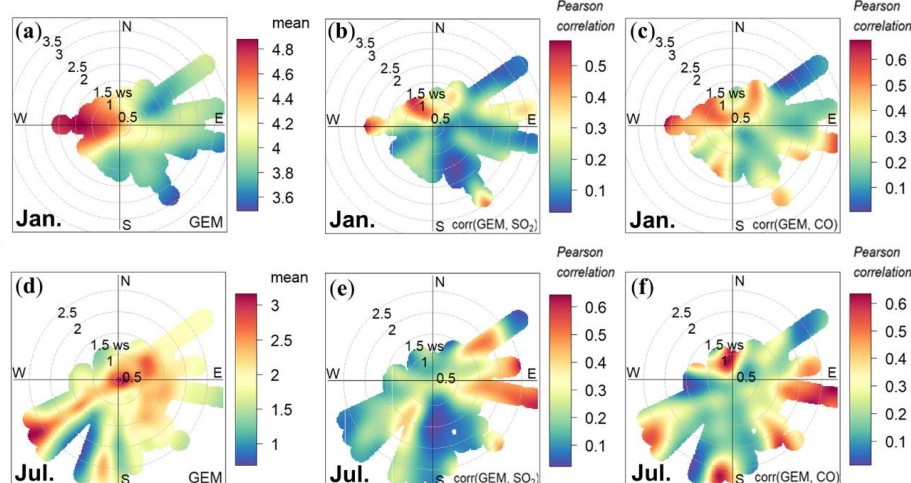

**Figure 4. Bivariate polar plots of GEM concentrations (a, d), the correlation coefficients of GEM with SO$_2$ (b, e) and CO (c, f) as a function of WS and WD in January and July during the whole study period. Note that GEM in ng m$^{-3}$, wind speed in m s$^{-1}$, wind direction in °.**

**3.2.2 Long distance migration**

The clusters of 72 h air mass backward trajectories and the potential source regions of GEM in January and July during the study years are shown in Fig. 5. On the whole, the GEM concentrations in Xiamen were under an influence of continental air masses in January and a mixing influence of continental and marine air masses in July. According to the direction of air mass trajectory clusters, the source regions of GEM in January were classified to the local region (Local), East China (EC), Southwest China (SWC), North China (NC) and Central China (CC). It can be seen that air masses from the Local and EC dominated in January, which accounted for 60.6% – 100% of the total trajectories over the study years. The GEM concentrations in Xiamen were also affected by the long-distance transport of air masses from Mongolia through the NC in January of 2013, 2015 and 2020 and from the SWC in January of 2015, 2017 and 2020. Table 2 summarizes the mean concentrations and weighted average contributions of GEM associating with different backward trajectory clusters in January and July. The GEM in January were mainly contributed by the Local and EC with the weighted average contributions more than 75% (except 2017 that was influenced by the CC). The results could reasonably be expected because anthropogenic Hg emissions in East China was extremely large due to the dense industries (Zhang et al., 2015). The North China Plain region was one of the heaviest mercury polluted area in China (Zhang et





al., 2015). But to our surprise, the GEM concentrations associated with air masses from the NC were
relatively low compared to air masses from other directions. This is most likely due to the effect of
transmission distances, while GEM concentrations decreased with long transport distances. In contrast,
short air mass trajectories, like from the Local, indicated a relatively stagnant air condition which is not
conducive to the diffusion of GEM (Zhang et al., 2021). The GEM concentrations of air masses from all
directions in 2015 were higher than those of other study years (Table 2). Previous studies have also
reported the high levels of GEM in 2015 among urban Beijing, Changdao Island, rural Shanghai and
Chongming Island (Tang et al., 2018; Qin et al., 2020; Wang et al., 2020; Wu et al., 2020), which suggests
a heavy GEM pollution on a large regional scale during 2015. The high GEM in January 2015 were most
likely linked to an adverse effect of meteorological conditions due to extreme 2015 – 2016 El Niño event,
which affected the levels of atmospheric pollutants directly via changing precipitation and large-scale air
circulation patterns or indirectly via impacting emissions (e.g. biomass burning) and re-emissions from
land/ocean (Monks et al., 2012; Carbone et al., 2016; Martin et al., 2017; Rowlinson et al., 2019; Yu et
al., 2019; Nguyen et al., 2022).

As shown in Fig. 5, the potential source regions of GEM in July were classified to Local, South China

Sea (SCS), Philippines Sea and Taiwan Strait (PhiS+TW), as well as Philippines Sea and East China Sea
(PhiS+ECS). The dominant clean marine air masses helped explain the lower concentration of GEM in
July than in January. The air masses arriving in Xiamen in July were mainly from the SCS and PhiS+TW,
accounting for more than 85% of the total trajectories (except 2015). The weighted average contributions
of GEM from the SCS were 21.4% in July 2015 and 36.1% in July 2017 (Table 2), however, the GEM
concentrations associating with the SCS were approximately 1.29 and 1.13 times higher than the average
GEM in July of 2015 and 2017. Thus, the elevated GEM concentrations of the SCS in July 2015 is due
to that the air masses passed more closely through Southeast Asia where the intense biomass burning
often occurred (Friedli et al., 2009; Sheu et al., 2013; Liu et al., 2016). A previous study on Hainan Island
also pointed to the possibility of long-range transport of GEM from Southeast Asia to South China (Liu
et al., 2016). Very differently, the high GEM of the SCS in July 2017 is likely because the air mass cluster
from the SCS was short which indicates a stagnant air condition, and the air mass cluster was close to
the land of Southeast China where were densely populated and highly industrialized (Yuan et al., 2021).





**Table 2. Concentrations (ng m⁻³) and weighted average contributions (%) of GEM associating with**
**different backward trajectory clusters in January and July over the study years.**

| Period | GEM | Source regions | | | | |
|---|---|---|---|---|---|---|
| | | Local | EC | NC | SWC | CC |
| **2012.01** | 3.50±0.62 | 3.21±0.62 (39.6%) | 3.72±0.53 (60.5%) | / | / | / |
| **2013.01** | 4.01±1.34 | 4.37±1.33 (30.3%) | 4.02±1.34 (63.5%) | 2.91±0.69 (6.0%) | / | / |
| **2015.01** | 4.47±0.78 | 5.05±0.96 (47.0%) | 4.31±0.50 (30.0%) | 4.32±0.54 (14.8%) | 3.74±0.26 (8.2%) | / |
| **2017.01** | 4.03±1.00 | 4.12±1.06 (45.1%) | 3.76±1.03 (15.1%) | / | 3.93±0.82 (10.0%) | 4.12±0.94 (29.9%) |
| **2020.01** | 3.93±0.88 | 4.18±1.01 (34.6%) | 4.00±0.81 (42.4%) | 3.43±0.64 (15.7%) | 3.71±0.69 (7.5%) | / |
| | | Local | SCS | PhiS+TW | PhiS+ECS | |
| **2012.07** | 2.49±0.79 | / | 2.83±0.85 (57.0%) | 2.14±0.63 (30.2%) | 2.19±0.41 (12.9%) | |
| **2013.07** | 2.18±0.65 | / | 2.11±0.67 (35.4%) | 2.23±0.66 (64.4%) | / | |
| **2015.07** | 2.55±0.86 | 2.38±0.77 (47.7%) | 3.30±0.68 (21.4%) | / | 2.18±0.56 (30.8%) | |
| **2017.07** | 2.65±0.69 | / | 3.00±0.81 (36.1%) | 2.47±0.54 (63.9%) | / | |
| **2020.07** | 1.56±0.55 | / | 1.54±0.54 (80.6%) | 1.75±0.62 (19.2%) | / | |



**Figure 5. Clusters of 72 h air mass backward trajectories in January (a, c, e, g, i) and July (b, d, f, h, j) during the study years. The legend represents the proportion of the trajectory after clustering and the color piece on behalf of the source regions.**



### 3.3 Factors affecting GEM concentrations

### 3.3.1 Model evaluation

GAMs were applied to investigate the influencing factors on the inter-annual trend of GEM in this study. The fitted (observed) GEM concentrations derived from the GAMs were approximately $4.00 \pm 0.45$ ng m$^{-3}$ ($4.00 \pm 0.84$ ng m$^{-3}$) in January and $2.23 \pm 0.33$ ng m$^{-3}$ ($2.23 \pm 0.61$ ng m$^{-3}$) in July, reflecting that the model approximates the concentration of GEM to the mean. The observed and fitted GEM concentrations showed a good consistency in time series (Fig. 6a) and the residuals were normally distributed (Fig. 6b). The $R^2$ of the observed and fitted GEM concentrations was 0.71 (Fig. S4) and the variance interpretation rate was 72.3%. In previous studies using GAMs to quantify impact factors on air pollutants, the $R^2$ was generally between 0.35 and 0.86 (Gong et al., 2018; Li et al., 2019; Wu et al., 2020; Wu et al., 2021; Zhang et al., 2021). The fitted result of the GAMs in our study falls in this range.

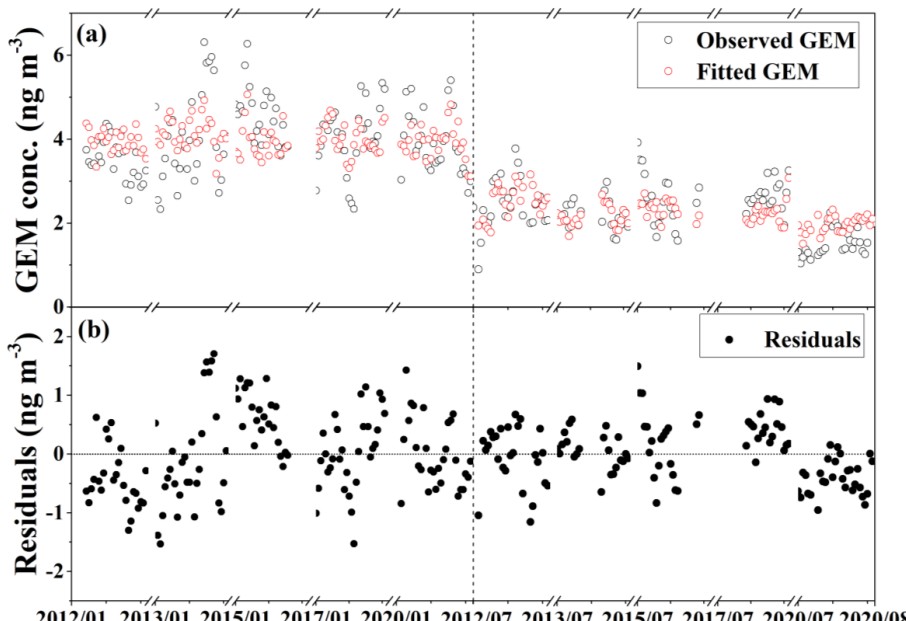

**Figure 6. Time series of (a) observed and fitted GEM concentrations in January and July and (b) residual distribution during the whole study period.**





### 3.3.2 Inter-annual variations of factor contributions

The GAMs screened out five variables, CO, RH and SP, as well as 24h-Latitude and 24h-Longitude, which represented three factors of anthropogenic emissions, meteorology and transmission, respectively. The contributions of the three factors to the variation of GEM concentrations in January and July of the study years are shown in Fig. 7. The inter-annual interpretation rate of anthropogenic emissions varied from 13.6% to 39.9% in January and from 15.9% to 49.6% in July, with a mean value of 26.9 ± 11.4% during the whole study period. The interpretation rate of the emission factor to the variation of GEM concentrations was pronouncedly high in 2012 and decreased overall to 2020, which reflects the effectiveness of emission mitigation measures in reducing GEM concentrations. China has adopted a series of desulphurization and denitrification measures for air pollution control during the period 2010 – 2015. The capacity of desulphurization and denitrification units in China had reached 99% and 92% of the total installed capacity of coal power plants, and $1.6 \times 10^8$ kw had been upgraded to achieve ultra-low emissions by 2015. In addition, strict restrictions of mercury on the mining, production, use, import and export were also be imposed in China since 2017 (http://www.mee.gov.cn, last access: 17 May 2022). We observed a positive relationship between GEM and variable CO in most of the CO observed range (Fig. 8a). That is, GEM concentrations basically decreased with the reduction of CO which stands for anthropogenic emissions. The most significant reduction in GEM concentrations explained by anthropogenic emissions was 0.70 ng m$^{-3}$ from July 2012 to July 2013 and 0.62 ng m$^{-3}$ from January 2013 to January 2015. After that, the effect of anthropogenic emissions on reducing GEM concentrations gradually diminished. The results indicate that, when the anthropogenic Hg emissions reach a relatively low level, the influence of emission reduction on GEM concentrations becomes less pronounced, and in turn, the influence of meteorology and transmission gradually become prominent.

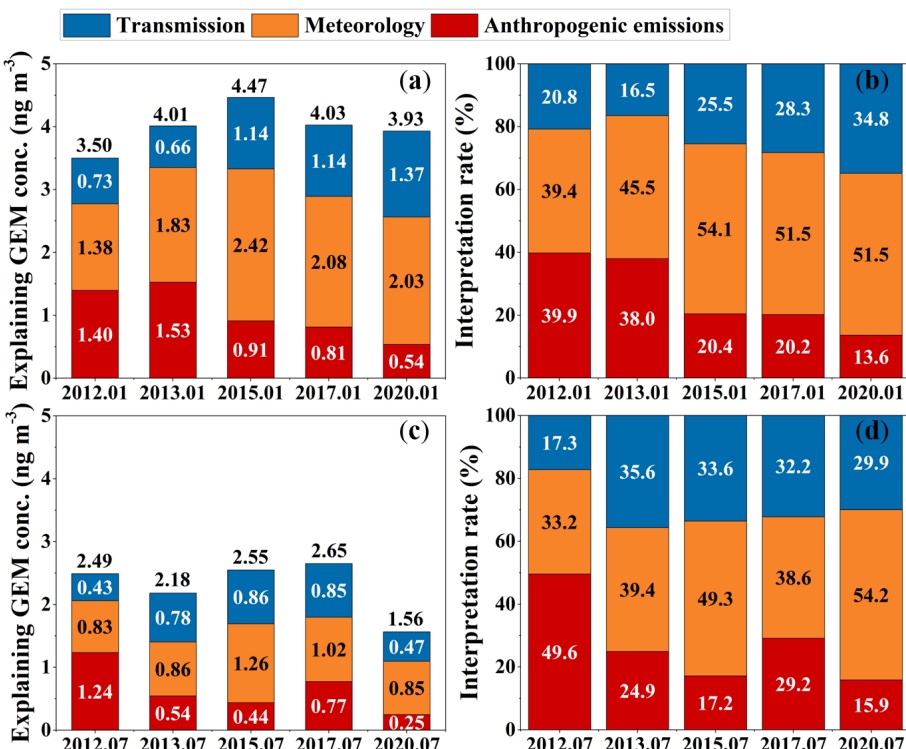

**Figure 7. The explaining concentration (ng m⁻³) (a, c) and interpretation rate (%) (b, d) of the three factors to the variation of GEM concentrations during the study years.**

Meteorology was the most important of the three factors, which contributed, on average, $45.7 \pm 7.2\%$ to the variation of GEM concentrations during the whole study period. The inter-annual interpretation rate of meteorology largely varied from 39.4% to 54.1% in January and from 33.2% to 54.2% in July. The largest part of GEM concentrations explained by meteorological factor occurred in 2015, which were 2.42 ng m⁻³ in January and 1.26 ng m⁻³ in July, respectively. According to the importance index F value (Table S2), RH and SP were the two most important parameters for the meteorological factor. As shown in Fig. 8b and 8c, GEM concentrations linearly increased with the increasing RH, while the relationship between GEM and SP was negative when SP < 990 hPa (mostly in July) and positive when SP > 1004 hPa (mostly in January). Thus, in January 2015, the RH and SP both had positive impacts on GEM concentrations, and we suspected that the highest SP (Table S5) could largely explain the high GEM concentrations in this period. The high interpretation rate of meteorological factor in July 2015 was mainly related to the remarkable high RH (82.4%) and low SP compared to the July of other years (Table

S5). The quite different meteorological parameters in 2015 from other years were likely driven by the
extreme 2015 – 2016 El Niño event, which led to the elevated GEM concentrations on a regional scale
as mentioned above.

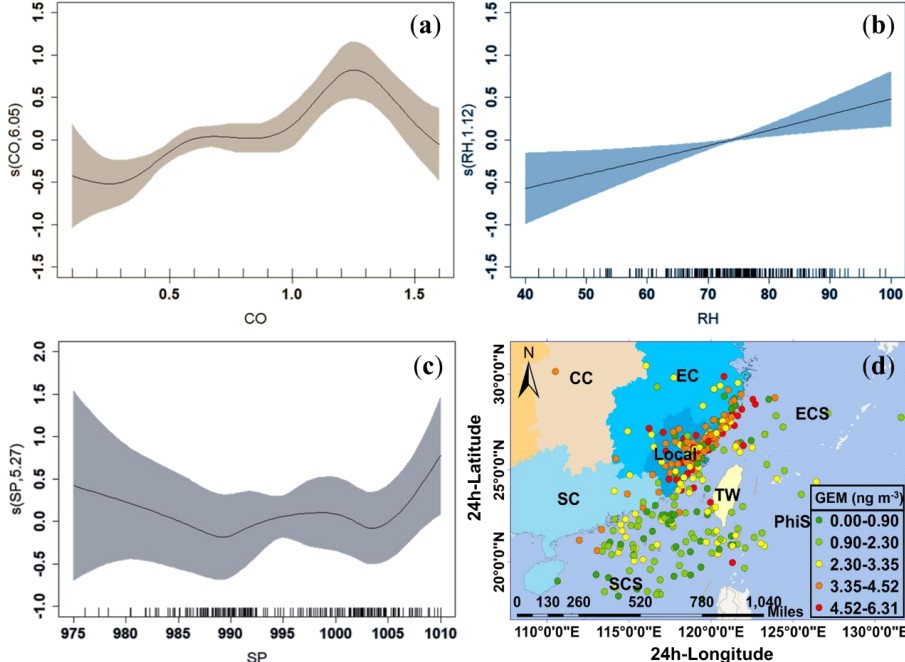


**Figure 8. Spline of GEM to the chosen variables, (a) CO (mg m$^{-3}$), (b) RH (%), (c) SP (hPa), and (d) 24h-Longtitude, 24h-Latitude. The grey background around the line is 95% confidence bounds for the response. The short lines on x axes show the distribution of data points. The number in the bracket of ordinate title is the estimated degree of freedom. The colored dots in figure (d) represents the interactive influence of 24h backward trajectories latitude and longitude coordinates.**

Regional transmission explained a proportion of GEM concentrations comparable to anthropogenic
emissions, which made up 27.4 ± 6.8% of total variation during the whole study period. During the period
of 2013 – 2020, the interpretation rate of transmission displayed an increasing trend from 16.5% to 34.8%
in January and a minor variation between 29.9% and 35.6% in July. The interaction of 24 h backward
trajectories location and corresponding GEM concentrations are shown in Fig. 8d. The high GEM
concentrations mostly presented in the areas with 24h-Longtitude of 116 – 124°E and 24h-Latitude of 24
– 30°N. This is consistent with the backward trajectory results in section 3.3.2 that the main source
regions of GEM were local area and East China. The interpretation rate of transmission increased





significantly in winter among years, indicating the growing importance of transmission from high Hg
emission zone to low emission zone in the background of anthropogenic emissions reduction. This
phenomenon well reflects the characteristics of GEM that undergoes long-range transport.
**4 Conclusions**
Long-term observation of GEM concentrations along with conventional pollutants and meteorological
parameters were conducted in Xiamen city, Southeast China. GEM concentrations showed no distinct
trends over the period 2012 – 2020. The temporal variation of GEM was characterized by higher values
in winter than in summer and in nighttime than in daytime, which is consistent with those in most urban
cities.
Local point sources contributed to the elevated GEM concentrations in individual seasons.
Nevertheless, the inter-annual variation trend of GEM was not consistent with those of local $SO_2$ and
$NO_x$ emissions, suggesting that anthropogenic emissions might be not the dominant influencing factor.
The trajectory results showed that the pronounced high GEM concentrations in winter 2015 was a
regional phenomenon, which was likely due to an adverse effect of meteorology due to extreme 2015 –
2016 El Niño event.
Three factors, i.e. anthropogenic emissions, meteorological conditions and transmission were
identified by GAMs, which explained 26.9 ± 11.4%, 45.7 ± 7.2% and 27.4 ± 6.8% to the variation of
GEM concentrations during the whole study period, respectively. Anthropogenic emissions showed a
decreasing interpretation rate since 2012, indicating the effectiveness of emission mitigation measures in
reducing GEM concentrations in the study region. In contrast, the interpretation rate of transmission in
winter displayed an increasing trend. Meteorology explained the largest proportion of GEM
concentrations, which was more likely the dominant factor influencing the inter-annual variation trend
of GEM in the study region.



**Data availability.** A dataset for this paper can be accessed at https://doi.org/10.5281/zenodo.6573605
(Shi et al., 2022). High-altitude meteorology parameters can be acquired from the European Centre for
Medium-Range Weather Forecasts (ECMWF) reanalysis (https://www.ecmwf.int, last access: last access:
23 March 2022), Gridded meteorological data are available from the Global Data Assimilation System
(https://www.arl.noaa.gov/, last access:5 May 2022), The details are also available upon request from the
corresponding author.

**Author contributions.** JS and LX designed this study and analysis the data. YuC, LX, YH, ML, XF,
YaC, CY GC, LT, JX and JC were involved in the scientific discussion and offered valuable suggestions
for modifications**.** JS and LX wrote the manuscript. YuC and JC helped revise the manuscript. and LY
managed finances. All authors reviewed the paper.

**Competing interests.** The authors declare that they have no conflict of interest.

**Disclaimer.** Publisher's note. Copernicus Publications remains neutral with regard to jurisdictional
claims in published maps and institutional affiliations.

**Acknowledgements.** We would like to thank the Xiamen Atmospheric Environment Observation and
Research Station of Fujian Province for providing data support for this research. We would like to thank
Siqing Zhang, for his help with the daily maintenance of the Tekran system. We also thank the members
of the Atmospheric Environment Research Group of the Urban Environment Institute for their help and
support in data analysis.

**Financial support.** This research was supported by National Natural Science Foundation of China (grant
no. 21507127; 41575146), the CAS Center for Excellence in Regional Atmospheric Environment (grant
no. E0L1B20201), and Xiamen Atmospheric Environment Observation and Research Station of Fujian
Province.



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
