# Peer review of "Measurement report: Atmospheric mercury in a coastal city of Southeast China: inter-annual variations and influencing factors"

_Atmospheric Chemistry and Physics, 2022_

## Author Comment (AC1)

The manuscript addresses the variability of atmospheric mercury concentration in a coastal city in Southeast China. The manuscript aims to report the main factors driving GEM variability by deploying the regression analysis method. The scientific question is relevant to the scientific community. However, many issues can be highlighted in the manuscript.

The main concern in the manuscript is its design, how the Generalized Additive Model was used, and the premises assumed for the pattern recognition of the factors driving GEM variability. The authors lack knowledge of the used method. The signal extracted from the matrix of trace gases, PM, and meteorological data used to reconstruct GEM, is not explicitly linked to GEM sources, transport, or processes. The factorization was constrained by a minimum concentration covariance that led to the meteorologic factor as the main cluster. I am afraid that the authors were misled by a spurious correlation in the propagation of the eigenvector, where the main factor explaining the GEM was seasonality. The main disadvantage of the unsupervised learning technique as the one used by the authors is the fact that the possible solution is no-unique.

Response: Thanks very much for your careful review and valuable comments which are very helpful for improving the quality of the manuscript. We have learned more GAM method through materials and literature. GAMs seem not like traditional unsupervised learning techniques such as PCA. We did not use GAMs to cluster, but to build a well-fitted nonlinear regression model and calculate the variation in factors interpretations rate. We carefully read the comments and revised the whole manuscript accordingly.

1. In this study, we assumed that the GEM concentrations were mainly affected by three factors: anthropogenic emissions, meteorology and transportation. 16 variables we obtained from site observation and web downloads were screened to represent the three factors using two methods: Statistical judgment and the meaning of the variables. The detailed screening processes have been implemented in the revised manuscript.

2. To eliminate the effect of seasonality on variance clustering, we have used "seasons" as an input variable when building the model with the whole dataset, and then run the model separately in summer and winter. We clarified this point in the **Section 2.5 Model establishment (Line 187-189)**. At the same time, we focused on inter-annual differences in individual seasons instead of seasonal comparisons when discussing the results.

3. We agree that it's better to use the variables which are explicitly linked to GEM sources, transport, or processes. However, we encountered some difficulties to obtain the high spatial resolution of Hg emissions inventory in China. In addition, there is no known explicit parameters to represent GEM processes in the atmosphere. The advantage of GAMs is that it can use routine monitoring or easily obtained parameters to represent the influencing factors. In revised manuscript, we explained the meaning of the retained variables in detail.

4. Yes, the possible solution of GAMs is no-unique. In this study, the accuracy of GAMs simulation was assessed using a 10-fold cross-validation test. The principle of the test is dividing the whole dataset into ten subsets randomly, and in each round of cross-validation, nine subsets are used to fit the model and the remaining one is predicted. This process is repeated 10 times to ensure that every subset is tested. The 10-fold cross-validation results showed a good coincidence between the GAMs and cross-validated result. In addition, we also use the gam.check function (e.g. Quantile-quantile (QQ) plots) to ensure the validity and

accuracy of the model. The detailed introduction could be found in **Section 2.5 Model Quality Control (Line 190-200).**

Specific comments:

Line 236: The authors call data from two months "trend over 2012"; however, it corresponds only to ten months of data for a period of nine years. The terminology "trend" is incorrect throughout the manuscript and should be revised. After all, it is not clear why the authors used only January and July data.

Response: Thanks for your suggestion. We used "variation" instead of "trend" in the revised manuscript. In addition, we added the explanation of the period of GEM observation data to the text **(Section 2.2, Observation period selection)**. The main reason was that the period of instrument malfunction was different among years. We used representative months of data so that the period of GEM data was consistent and the GEM data could be comparable among years. We chose January and July data mainly based on two considerations: (1) The measurement site, Xiamen, is located in the coastal region of Southeast China under the control of East Asian monsoon, which has a significant distinction in meteorology between summer and winter; (2) Based on our previous study on GEM observations in Xiamen throughout a year (Xu et al., 2015), January is very representative of winter and July is representative of summer.

Line 239-249: The emission data should be presented, and regression with observation should be discussed.

Response: It's a pity that we do not have the Hg emission inventories data. We summarized the published data of anthropogenic Hg emission in China so far, and added it to the supporting information **(Figure S3,4)**. The published data did not cover the study period 2012-2020, and the small amount of annual emission data might be not suitable for regression analysis. According to the published data, Wu et al. (2016) estimated atmospheric Hg emissions in China decreasing from 547 tons in 2010 to 530 tons in 2014. The report from AMAP/UNEP showed that the anthropogenic Hg emissions in China were 565.2 t in 2015 relative to 575.2 t in 2010. An inventory over the period 1978-2017 revealed that China's anthropogenic Hg emission was highest in 2013 and then decreased until 2017 (Liu et al., 2019). It could be expected that the anthropogenic Hg emissions in China had a downward trend over the period 2012-2020 and the peak emission was most likely to occur in 2012 to 2014. A more detailed description of Hg emissions was added in the **Section 3.1.1 (Line 225-231)**.

[Figure]

Figure S3. Anthropogenic mercury emissions from China reported in the literature (Streets et al., 2005; Wu et al., 2006; Cheng et al., 2015; Tian et al., 2015; Zhang et al., 2015; Wu et al., 2016; Liu et al., 2019).

[Figure]

Figure S4. Anthropogenic mercury emissions during 1978-2017 in China (Liu et al., 2019).

Line 243: "aggressive" what does it mean?

Response: We mean "vigorous measures". In 2013, the Chinese State council issued an air pollution prevention and control action plan. Since then, plenty of emissions control measures, like accelerate the elimination of backward production capacity, accelerate the promotion of central heating, upgrades and building air pollution control devices, have been widely implemented in China (**Line 244-247**). We changed "aggressive" to "vigorous measures" in the revise version.

Line 252: Would it be possible to show the coal consumption in Fujian and China?

Response: As you suggested, we provided coal consumption in Fujian and China during 2012-2020 both in **Fig. S5**. The data came from the Statistical Yearbook of China and Fujian (http://www.stats.gov.cn/tjsj/ndsj/: last access: 15 June 2022). The coal consumption in Fujian and China exhibited a similar variation, firstly decreasing to a valley in 2016 and then showing an upward trend from 2016 to 2020 (**Fig. S5**).

[Figure]

Figure S5 Statistics of annual coal consumption in China and Fujian Province during 2012 – 2020.

Line 259: Probably, the authors mean inter-annual variation rather than an inter-annual trend. I am afraid that the data exploitation presented by the authors does not allow a proper evaluation of the trend.
Response: We agree with you. We used "inter-annual variation" instead of "inter-annual trend" in the text and focused on inter-annual comparison of GEM concentration when revising the manuscript.

Section 3.1.2

I am afraid that using only two months is inappropriate for seasonality evaluation. In addition, one month represents only 1/3 of the season.
Perhaps it would be more appropriate to call the section January/July comparison rather than "seasonal".
Response: According to the climate features in the study region, January and July could well reflect the characteristic of the winter and summer seasons, respectively. The observation site Xiamen is located in the coastal region of Southeast China under the control of subtropical oceanic monsoon, which has a significant distinction in meteorology between summer and winter. In addition, a whole year GEM concentration observation in Xiamen supported that the GEM data in January can represent the GEM characteristics of winter and July can represent summer (Xu et al., 2015). We added the instruction of the season's representation of January and July in **Section 2.2 Observation period selection**.

Line 271-282: The polar plot does not support the statement of dominant wind from the North or a higher concentration of GEM on this wind. If the plots are correct, the predominant source of GEM in January is in the west, and long transport does not play a major role in the level of GEM at Xiamen. Actually, the plot shows only a low level of GEM at wind from the sea.

Response: We are sorry we did not explain the air mass and near-ground wind directions clearly. The wind speed/direction used in the polar plot analysis was observed near the ground and represents a very local scale (within Xiamen city) airflow condition. The wind speed/direction was strongly affected by the terrain. The "wind" here refers to the air masses which mainly affected by large-scale atmospheric circulation. The air masses in the study region was mostly originated from the land area in winter and from the ocean in summer. To be clearer, we've changed "wind" here to "air mass".

Line 283-288: It seems confusing; the authors should consider rewording it.
Response: We rewrote this passage as follows: "The diurnal variations of bihourly GEM concentrations were consistent among years (Fig. 3). In general, the GEM concentration peaked in the early morning, decreased to a valley in the afternoon, and then rose during the night. The diurnal pattern of GEM concentrations in January 2015 were gentle than other years of the same period, which might be related to the enhanced effect of air mass transport (Fu et al., 2012; Nguyen et al., 2022)". **(Line 277-281)**

Line 289: The diurnal pattern observed for July can be potentially constrained by sea/land breeze since it is a coastal place.
Response: We agree that the sea land breeze (SLB) is a potential factor of atmospheric mercury concentrations in coastal cities. Our statistics for 2017 – 2020 show that the average number of SLB days was only 6 days in January and 3 days in July. In addition, we compared the diurnal trend of atmospheric mercury in January and July of 2017 and 2020 on the SLB days and non-SLB days. As shown in the chart below, there was no significant difference in the diurnal pattern of GEM between SLB and non-SLB days. Thus, we infer that sea land breeze was not the dominant factor influencing the diurnal trend of mercury concentrations.

[Figure]

Line 297 – 298: For kinetic reasons, photo-oxidation cannot be the explanation for the observed reduction of GEM in the daytime. It is most like related to GEM fluxes. The authors speculate about the diurnal variation of GEM without a solid clue about the processes driving it.
Response: We agree with your comment. The photo-oxidation of GEM possible contributed a small part to the diurnal variation of GEM, but it was not the dominant factor for the daytime

GEM reduction in the study region. We clarified this point in the revised manuscript **(Line 289-291)**.

Polar plots are quite limited in providing emission locations. Concentration-Weighted Trajectory could improve this section; it would map GEM, allowing hotspot concentration identification.

Figure 5 does not bring insight into the mercury source location. A different kind of plot should be presented. In addition, a clearer CWT method should be presented.

Response: The polar plots analysis here was used for the identification of local point sources mainly within Xiamen city. According to your suggestion, we further performed CWT analysis and revised the relevant discussion in the main text **(Section 3.2.2).**

[Figure]

Line 365-370: It seems a last-minute explanation; since only fluxes can explain variation in the atmospheric mercury concentration, the authors should look into Hg emission to address a more convincing explanation.

Response: We rewrote this section and considered primarily the impact of Hg emissions, and secondarily the impact of meteorology **(Line 357-363)**.

"The high GEM concentrations in January 2015 was likely due to a combination of high-level Hg emissions and adverse meteorological conditions. The annual atmospheric mercury emissions in China were about 565 tons in 2015, which was roughly 20% higher than they were in 2010 (AMAP/UNEP, 2018). According to anthropogenic mercury emissions inventory in China during 1978 – 2017, mercury emissions might peak around 2013, and remained high in 2015 (Figure S2). In addition, an adverse effect of meteorological conditions due to extreme 2015 – 2016 El Niño event might result in an increase of GEM concentrations in 2015 (Nguyen et al., 2022)."

Section 3.3.2

This section has major concerns

Unsupervised learning techniques are power statistic methods applied successfully to extract signal and meaning information from high-dimensional data. Deploying nonnegative matrix factorization, we can more than explain covariance; we can extract the pattern of source and transport of atmospheric trace gases. However, I am afraid that the authors did not design the factorization properly. The species considered in the matrix were chosen without criterium. It was only convenient for the authors to have those species there. What is the sense of having PM in the matrix? Considering species that do not bring retrieval signals will not provide insight into mercury processes/source/fade. It only increases uncertainty and chances of spurious correlation, misleading the eigenvector's propagation.

Response: Thanks very much for your valuable comment. We added a detailed introduction of GEM factor selection into **Section 2.5 Parameter Selection** in the revised manuscript.

1, 16 variables we obtained from site observation and web downloads generally fell in four categories: anthropogenic emissions ($SO_2$, $NO_2$, $O_3$, CO, $PM_{2.5}$, $PM_{10}$), surface meteorology (T, RH, WS, WD, SP), high-altitude meteorology (BLH, UVB and LCC) and air transmission transportation (24h-Latitude and 24h-Longitude). All the variables were standardized by min-maximum method. The normalized data eliminates the effects of differences in dimension and ranges of values between indicators. The standardized variables were then screened using two methods: Statistical judgment and the meaning of the variables.

2, The detailed processes of factor screening are as follows. 1) We performed collinearity diagnostics with all the parameters. $PM_{10}$, $SO_2$, $NO_2$, SP and UVB were rejected into the model due to their high collinearity (VIF > 5). 2) we considered the meaning of the remaining parameters based on the literature and our research experience. CO is mainly sourced from anthropogenic emissions and has a long atmospheric residence time (compared to $SO_2$ and $NO_2$). In addition, Hg emissions in Fujian provinces were dominated by combustion sources (Liu et al., 2019). Hence, we used CO to represent anthropogenic Hg emissions. Parameters $O_3$ and $PM_{2.5}$ were easily rejected into the model. 3) After determining the first parameter CO, we put the remaining parameters (WS, WD, T, RH, BLH, LCC, 24h-Latitude and 24h-Longitude) into the model one by one. As the parameters were successively added into the model, the AIC decreased and $R^2$ increased. In this step, WS, WD and LCC were rejected. Eventually, 6 variables including CO, RH, T, BLH, 24h-Latitude and 24h-Longitude were selected into the model.

3, Considering that parameters of the same category may interact, we used interaction functions of tensors. RH, T and BLH interaction was used to represent the meteorological factor. 24h-Latitude and 24h-Longitude interaction was used to represent the transportation factor. Given that the 6 selected variables passed the collinearity test, the three factors they represented, i.e., anthropogenic Hg emissions, meteorological and transportation were considered to be independent of each other.

The major problem in this study was the correlations extracted from the species inserted in the factorization. The differences in the GEM concentration through the season, which are dependent on the seasonality of the emissions, were correlated with the seasonality of the meteorological parameters, which were extracted as the causes of GEM reduction in July. The direct incorporation of meteorological parameters into the factorization misled the eigenvector propagation. The seasonal differences created cluster minimizing the variance but do not sign origin/source/or fluxes of GEM.

The factor obtained by the authors does not provide any insight into the GEM reducibility (computationally speaking) since it does not bring information on the source/or fluxes of GEM. Seems that the authors did not plan the species to be considered in the calculation.

Moreover, the meteorological variables cannot be included directly in the factorization matrix. In order to evaluate transport, the authors should use an inversion accoupled with a transport model.

I hope the authors do not feel disappointed or frustrated with my comments. I`m very enthusiastic about unsupervised learning methods for pattern recognition and estimation of fluxes and the implementation of nonnegative matrix factorization into inversion modeling. Indeed, it has great potential to bring new insight into atmospheric mercury reducibility. I hope the authors only feel motivated to learn and improve their research.

Response: We really appreciate your comments and suggestions. We learned a lot from your comments. The inversion model is indeed a very meaningful work for regional and global flux estimation and has great potential to bring new insight into atmospheric mercury reducibility. Whereas, the main purpose of this study was to recognize the factors driving the inter-annual variations of GEM concentrations, which is a little different from building an inversion model of atmospheric GEM. Some explanations are listed as follows:

1, We agree with that the seasonal differences may create cluster minimizing the variance. We did take seasons into account. When we built the model with whole dataset, we have used "seasons" as an input variable. and then run the model separately in summer and winter. The accuracy of GAMs simulation was assessed using a 10-fold cross-validation test. We clarified this point in the revised manuscript **(Line 187-192).** As for the results discussion, we focused on inter-annual differences in individual seasons instead of seasonal comparisons.

2, According to your suggestion, we used the min-maximum method to standardize the parameters, including the meteorological variables, before the model running. The normalized data eliminates the effects of differences in dimension and ranges of values between indicators.

3, We have thought over the Hg species and from the characteristics of Hg species and the available representative variables **(Line 97-104).** GEM has a low chemical reactivity comparing to GOM and PBM in the atmosphere. GEM concentrations are largely affected by factors like anthropogenic emissions and atmospheric physical processes, which could be well represented

by routine monitoring or easily obtained parameters, like CO, RH, BLH, etc. Whereas, GOM and PBM concentrations were strongly affected by chemical transformation processes, which have no known suitable indicators. In addition, involving chemical transformation effect in the GAM models would make it more complicate.

4, We have not done studies on the inversion model. We agree that it has great potential to bring new insight into atmospheric mercury reducibility. We will learn more about the inversion model and we are very willing to provide the atmospheric mercury observation data from the two coastal cities of China for the verification of the inversion model.

**References**

Cheng, K., Wang, Y., Tian, H. Z., Gao, X., Zhang, Y. X., Wu, X. C., Zhu, C. Y., and Gao, J. J.: Atmospheric Emission Characteristics and Control Policies of Five Precedent-Controlled Toxic Heavy Metals from Anthropogenic Sources in China, Environmental Science & Technology, 49, 1206-1214, https://doi.org/10.1021/es5037332, 2015.

Fu, X. W., Feng, X., Liang, P., Deliger, Zhang, H., Ji, J., and Liu, P.: Temporal trend and sources of speciated atmospheric mercury at Waliguan GAW station, Northwestern China, Atmospheric Chemistry and Physics, 12, 1951-1964, https://doi.org/10.5194/acp-12-1951-2012, 2012.

Liu, K., Wu, Q., Wang, L., Wang, S., Liu, T., Ding, D., Tang, Y., Li, G., Tian, H., Duan, L., Wang, X., Fu, X., Feng, X., and Hao, J.: Measure-Specific Effectiveness of Air Pollution Control on China's Atmospheric Mercury Concentration and Deposition during 2013-2017, Environ Sci Technol, 53, 8938-8946, https://doi.org/10.1021/acs.est.9b02428, 2019.

Nguyen, L. S. P., Nguyen, K. T., Griffith, S. M., Sheu, G. R., Yen, M. C., Chang, S. C., and Lin, N. H.: Multiscale Temporal Variations of Atmospheric Mercury Distinguished by the Hilbert-Huang Transform Analysis Reveals Multiple El Nino-Southern Oscillation Links, Environ Sci Technol, 56, 1423-1432, https://doi.org/10.1021/acs.est.1c03819, 2022.

Streets, D., Hao, J., Wu, Y., Jiang, J., Chan, M., Tian, H., and Feng, X.: Anthropogenic mercury emissions in China, Atmospheric Environment, 39, 7789-7806, https://doi.org/10.1016/j.atmosenv.2005.08.029, 2005.

Tian, H. Z., Zhu, C. Y., Gao, J. J., Cheng, K., Hao, J. M., Wang, K., Hua, S. B., Wang, Y., and Zhou, J. R.: Quantitative assessment of atmospheric emissions of toxic heavy metals from anthropogenic sources in China: historical trend, spatial distribution, uncertainties, and control policies, Atmospheric Chemistry and Physics, 15, 10127-10147, https://doi.org/10.5194/acp-15-10127-2015, 2015.

Wu, Q. R., Wang, S. X., Li, G. L., Liang, S., Lin, C. J., Wang, Y. F., Cai, S. Y., Liu, K. Y., and Hao, J. M.: Temporal Trend and Spatial Distribution of Speciated Atmospheric Mercury Emissions in China During 1978-2014, Environmental Science & Technology, 50, 13428-13435, https://doi.org/10.1021/acs.est.6b04308, 2016.

Wu, Y., Wang, S., Streets, D. G., Hao, J., Chan, M., and Jiang, J.: Trends in anthropogenic mercury emissions in China from 1995 to 2003, Environ Sci Technol, 40, 5312-5318, https://doi.org/10.1021/es060406x, 2006.

Xu, L., Chen, J., Yang, L., Niu, Z., Tong, L., Yin, L., and Chen, Y.: Characteristics and sources of atmospheric mercury speciation in a coastal city, Xiamen, China, Chemosphere, 119, 530-539, https://doi.org/10.1016/j.chemosphere.2014.07.024, 2015.

Zhang, L., Wang, S., Wang, L., Wu, Y., Duan, L., Wu, Q., Wang, F., Yang, M., Yang, H., Hao, J., and Liu, X.: Updated Emission Inventories for Speciated Atmospheric Mercury from Anthropogenic Sources in China, Environmental Science & Technology, 49, 3185-3194, https://doi.org/10.1021/es504840m, 2015.

---

## Author Comment (AC2)

The manuscript by Shi et al. measured GEM concentrations in January and July in five individual years from 2012 to 2020, and the data were used to explore the potential factors controlling the inter-annual variations. Long-term measurements of GEM concentration are a useful tool for assessing of controls of regional anthropogenic emissions and global changes, and thus the data presented here are valuable. In the present study, the authors combined the multiple approaches including the analysis of GEM concentrations, criteria pollutants, backward trajectory and generalized additive model. I agree that it is practicable and relevant for using these kinds of methods to explore the controls in the change of atmospheric GEM. The manuscript is overall well organized, and can be read easily. I broadly agree with the discussions and findings of this manuscript. I therefore suggest a minor to moderate revision of this manuscript before final publication in ACP.

Response: Thanks very much for your overall positive review and valuable comments, which are very helpful for improving the quality of our manuscript.

As mentioned at the beginning of the manuscript, the major objectives of long-term observations are to evaluate the changes in anthropogenic emissions, that is an important part for the implementation of the Minamata Convention on Hg. However, after a comprehensive analysis, the authors mostly highlight that the changes in meteorological conditions were the most important variable in controlling the long-term trend in GEM. This is valuable, but not very striking findings to me because it is well accepted that variations in GEM among different short periods (e.g., monthly) could be impacted by changing atmospheric transport (air transport would change with different periods and subsequently affect the source-receptor relationships). Thus, I would suggest the authors to focus on the impact of changing local and regional anthropogenic emissions and climate on the trends in GEM concentrations, which would better serve for their research objectives.

Response: According to your suggestions, we summarized the published data of China's Hg emissions inventory so far **(Figure S3, S4)**, and placed greater emphasis on the impact of changing local and regional anthropogenic emissions and climate on the trends in GEM concentrations in the whole text as well as in the summary and conclusions **(e.g. Line 221-231, Line 357-363)**.

I am not clear why the meteorology is the major divers of changing GEM concentrations, and it also difficult to differentiate the impacts of meteorology, transmission, and emissions. I suspect that the transmission should be related to meteorology because the changes in local and regional meteorological conditions would further affect the transmission. Would the meteorology change land surface emissions and or atmospheric reactions that further affect the GEM? In addition, several previous studies reported declines in GEM concentrations in eastern China. Would this be an important cause for the changing contributions from transmission and meteorology? Thus, the authors may better define the three factors, which would help to better understand the real causes for the changes in GEM concentrations.

Response: Thanks very much for raising this critical issue. Yes, regional meteorological conditions and anthropogenic emission amount would affect transportation, and meteorological conditions also impacted land surface emissions and atmospheric reactions. Actually, it is impossible to differentiate the impacts of meteorology, transmission, and emissions in a realistic

scenario. Statistical methods simplify this realistic scenario and use representative parameters to indicate the factors.

The selection of representative variables for the factors was very crucial and decided what the factors mean. 16 variables we obtained from site observation and web downloads were screened by collinearity test, considering the meaning of variables, and statistical judgement. As a result, CO was used to represent the anthropogenic emissions factor; RH, T and BLH interaction was used to represent the meteorological factor; and 24h-Longtitude, 24h-Latitude interaction was used to represent the transportation factor. Given that the 6 selected variables passed the collinearity test, the three factors were considered to be independent of each other. We have added a detailed explanation of the factor's selection and the meanings of the factors in the revised manuscript. The detailed responses to this issue could be seen below.

Line 144-145: why did the authors only conduct two-month observations at the sampling site? Why not conduct a year of continuous observations for the selected years? A two-month observations in one year are sometime not adequate for assessing the inter-annual variations because of many factors mentioned in the manuscript.

Response: The main reason was that the period of instrument malfunction was different among years. For example, data for October, November and December were missing in 2013 and data for August, September and December were missing in 2015. We used representative months of data so that the period of GEM data was consistent and the GEM data could be comparable among years. We used January and July data mainly based on two considerations: (1) The measurement site, Xiamen, is located in the coastal region of Southeast China under the control of East Asian monsoon, which has a significant distinction in meteorology between summer and winter; (2) Based on our previous study on GEM observations in Xiamen throughout a year (Xu et al., 2015), January is very representative of winter and July is representative of summer. We added a description of seasonal representation to **Section 2.2 Observation period selection (Line 106-114)**. Furthermore, considering the influence of gap years, we have used "variation in winter/summer among years" instead of "inter-annual trend" in the revised manuscript.

Line 166-167: the definition of local impact relating to air mass within a province might over-estimate the local effect. Why not define the local impact within the city?

Response: In this study, polar plot was used to analyze the influence of local sources within Xiamen City. As for the regional source identification, we found that the endpoint of 72h backward air mass trajectories was relatively rare within Xiamen city. "Local" in air mass trajectory analysis means the emission source surrounding the site (mostly within Fujian province) compared to regional transportation. We have modified the backward trajectory analysis to a CWT analysis and changed "Local" in regional sources regions to "Fujian province" in the revised manuscript (**Figure 5 and Section 3.2.2**), which can display the source region more clearly.

Line 198-204: CO is mainly sourced from anthropogenic emissions but has a long atmospheric residence time, it may therefore a best proxy of local anthropogenic emissions. I would suggest the authors to consider using $SO_2$, $NO_2$, or $PM_{10}$ to define the anthropogenic factor, although these parameters would have relative weaker correlations with GEM. Why use RH and SP to

define meteorology? How could these two factors affect GEM concentrations? What are the 24h-latitude and -longitude? Are they referred as the air massed originated outside the city to define long-range transport?

Response: Thanks very much for the valuable comment. We added a detailed description of the variables screening into **Section 2.5 Parameter Selection** and some factor explanations into the corresponding discussion in **Section 3.3.2**.

(1) CO is mainly sourced from anthropogenic emissions and has a long atmospheric residence time (compared to $SO_2$ and $NO_2$). In addition, Hg emissions in Fujian provinces were dominated by combustion sources (Liu et al., 2019). Hence, we used CO to represent anthropogenic Hg emissions.

(2) According to another reviewer's comment, we used the min-maximum method to standardize the parameters before the model running. SP was rejected by collinearity test. We filtered remaining meteorological parameters based on the performance of the model fit as well as the variables importance. In this step, WS, WD and LCC were rejected and T, RH, and BLH were retained. The meteorological factor was represented by T, RH, and BLH interaction. The meteorological factor influenced GEM variations by several pathways. To better illustrate how the parameters affect GEM concentrations, we plotted the corresponding curves of GEM with variables (**Fig. 8**). GEM concentrations increased with the raise of temperature, possibly because the increase of temperature would promote the Hg emissions from natural surface. GEM concentrations increased with the raise of RH, possibly because the high RH environment was favorable for the liquid phase reduction of reactive Hg to GEM (Horowitz et al., 2017; Saiz-Lopez et al., 2018; Huang et al., 2019). BLH primarily indicated the physical status of the atmosphere especially diffusion conditions. We also gave a case to the combined effect of T, RH, and BLH on GEM variations (**Fig. S8**).

(3) The interaction of 24h-Latitude and 24h-Longitude was defined as the regional air mass transport, which represented the backward trajectory endpoint location during last 24 h.

Line223: the range of background GEM concentrations of 1.5-1.7 ng m$^{-3}$ is somewhat higher to me. Better to use recent global observations.

Response: The most recent data we found was the GEM concentrations at ground-based Global Mercury Observation System (GMOS) sites in 2013 and 2014. In the northern hemisphere, GEM measurements for background sites were ~1.5 ng m$^{-3}$ (Sprovieri et al., 2016; Diéguez et al., 2019). We have updated the data in the manuscript **(Line 207)**.

Figure 2: a statistic of the annual GEM concentrations should be added

Response: It might be not very suitable to do a statistic of the annual GEM concentrations in this study, because we used only two representative months of GEM data. To be more accuracy, we have compared the Jan. and Jul. data with the corresponding winter and summer data from the references rather than annual average data (**Line 210-216**).

Line 274: the GEM lifetime here is not consistent with that in line 76

Response: We finally unified the GEM lifetime to 0.5~2 years (Schroeder and Munthe, 1998).

Line 297-298: elevated O₃ and decreasing GEM concentrations should be mainly related to subsidence of free troposphere and daytime production of O₃. If the daytime declines in GEM is caused by oxidation, we would expect a much higher oxidation rate than experimental studies.

Response: We agree with you. We have clarified that the oxidation of GEM might contribute a small part to the daytime reduction of GEM, but was not the driving factor due to its low oxidation rate. We have highlighted the impact of subsidence of free troposphere in the revised manuscript **(Line 289-291)**.

Line 317: a citation of references should be added here

Response: We added the reference (Air pollution emissions inventory of Xiamen city in 2016, unpublished report) here (**Line 309-311**).

As shown in Figure 5: a large fraction of air masses originated or passed over oceans, please add their weighted GEM concentrations in Table 2

Response: Thanks for your suggestion. In the revised manuscript, we have further performed CWT analysis based on backward air mass results, which brings a better insight into the mercury source locations. The results of CWT analysis are showed in below figures. The corresponding discussion was also revised.

[Figure]

Line 424-426: it is difficult to expect low regional anthropogenic emissions because the GEM measured are much higher the background levels in East Asia. I suspect that the other two factors of transmission and meteorology were also impacted by changing local and regional anthropogenic emissions. Actually, I do not know what are these three factors representing. Are the anthropogenic emission and transmissions representing local anthropogenic contributions and regional background? What is the meteorology? Is it representing natural emissions'?

Response: We agree with that the regional high GEM concentrations affected the GEM in the study region, and the other two factors of transmission and meteorology were also impacted by changing local and regional anthropogenic emissions. Actually, it is impossible to differentiate the impacts of meteorology, transmission, and emissions in a realistic scenario. As above responses mentioned, statistical methods simplified this realistic scenario and used representative parameters to indicate the factors. Thus, the representative variables for the factors mostly decided what the three factors represent. 6 variables including CO, RH, T, BLH, 24h-Latitude and 24h-Longitude were screened out to represent the factors. We have added a detailed explanation of the factor's selection (**Section 2.5**) and the meanings of the factors in the corresponding discussion (**Section 3.3.2**)

(1) The anthropogenic emissions factor was represented by CO. CO and GEM have a long lifetime in the atmosphere, and anthropogenic Hg emissions in Fujian were dominated by combustion sources (Liu et al., 2019), so we used CO to represent anthropogenic Hg emissions.

(2) The transportation factor was represented by 24h-Longtitude, 24h-Latitude interaction. 24-latitude and 24-longtitude referred to the direction and distance of the 24h backward air mass from the study site. We did not put the regional background concentration as a parameter into the model. But this factor (the interaction of 24-latitude and 24-longtitude) would cover the effect of regional background of GEM concentrations.

(3) The meteorological factor was represented by RH, T and BLH interaction. The meteorological factors affected GEM concentrations by several pathways: (a) Hg chemical transformation. For example, RH was highly related to the liquid phase reduction of reactive Hg in the atmosphere (Horowitz et al., 2017; Saiz-Lopez et al., 2018; Huang et al., 2019). (b) Hg emissions from the natural surfaces. Substantial flux measurements have clearly showed that the elevated temperature and solar radiation can significantly promote the Hg re-emissions from these nature surfaces. (c) physical state of the atmosphere. A higher boundary layer likely indicated good diffusion conditions and was more conducive to the reduction of pollutant concentrations.

In order to better illustrate the meteorological influence, we added a typical case for additional clarification (**Figure S8, Section 3.3.2**).

**References**

Diéguez, M. C., Bencardino, M., García, P. E., D'Amore, F., Castagna, J., De Simone, F., Soto Cárdenas, C., Ribeiro Guevara, S., Pirrone, N., and Sprovieri, F.: A multi-year record of atmospheric mercury species at a background mountain station in Andean Patagonia (Argentina): Temporal trends and meteorological influence, Atmospheric Environment, 214, https://doi.org/10.1016/j.atmosenv.2019.116819, 2019.

Horowitz, H. M., Jacob, D. J., Zhang, Y., Dibble, T. S., Slemr, F., Amos, H. M., Schmidt, J. A., Corbitt, E. S., Marais, E. A., and Sunderland, E. M.: A new mechanism for atmospheric mercury redox chemistry: implications for the global mercury budget, Atmospheric Chemistry and Physics, 17, 6353-6371, https://doi.org/10.5194/acp-17-6353-2017, 2017.

Huang, Q., Chen, J., Huang, W., Reinfelder, J. R., Fu, P., Yuan, S., Wang, Z., Yuan, W., Cai, H., Ren, H., Sun, Y., and He, L.: Diel variation in mercury stable isotope ratios records photoreduction of PM2.5-bound mercury, Atmospheric Chemistry and Physics, 19, 315-325, https://doi.org/10.5194/acp-19-315-2019, 2019.

Liu, K., Wu, Q., Wang, L., Wang, S., Liu, T., Ding, D., Tang, Y., Li, G., Tian, H., Duan, L., Wang, X., Fu, X., Feng, X., and Hao, J.: Measure-Specific Effectiveness of Air Pollution Control on China's Atmospheric Mercury Concentration and Deposition during 2013-2017, Environ Sci Technol, 53, 8938-8946, https://doi.org/10.1021/acs.est.9b02428, 2019.

Saiz-Lopez, A., Sitkiewicz, S. P., Roca-Sanjuan, D., Oliva-Enrich, J. M., Davalos, J. Z., Notario, R., Jiskra, M., Xu, Y., Wang, F., Thackray, C. P., Sunderland, E. M., Jacob, D. J., Travnikov, O., Cuevas, C. A., Acuna, A. U., Rivero, D., Plane, J. M. C., Kinnison, D. E., and Sonke, J. E.: Photoreduction of gaseous oxidized mercury changes global atmospheric mercury speciation, transport and deposition, Nat Commun, 9, 4796, https://doi.org/10.1038/s41467-018-07075-3, 2018.

Schroeder, W. H. and Munthe, J.: Atmospheric mercury - An overview, Atmospheric Environment, 32, 809-822, https://doi.org/10.1016/S1352-2310(97)00293-8, 1998.

Sprovieri, F., Pirrone, N., Bencardino, M., D'Amore, F., Carbone, F., Cinnirella, S., Mannarino, V., Landis, M., Ebinghaus, R., Weigelt, A., Brunke, E.-G., Labuschagne, C., Martin, L., Munthe, J., Wangberg, I., Artaxo, P., Morais, F., Jorge Barbosa, H. d. M., Brito, J., Cairns, W., Barbante, C., del Carmen Dieguez, M., Elizabeth Garcia, P., Dommergue, A., Angot, H., Magand, O., Skov, H., Horvat, M., Kotnik, J., Read, K. A., Neves, L. M., Gawlik, B. M., Sena, F., Mashyanov, N., Obolkin, V., Wip, D., Bin Feng, X., Zhang, H., Fu, X., Ramachandran, R., Cossa, D., Knoery, J., Marusczak, N., Nerentorp, M., and Norstrom, C.: Atmospheric mercury concentrations observed at ground-based monitoring sites globally distributed in the framework of the GMOS network, Atmospheric Chemistry and Physics, 16, 11915-11935, https://doi.org/10.5194/acp-16-11915-2016, 2016.

Xu, L., Chen, J., Yang, L., Niu, Z., Tong, L., Yin, L., and Chen, Y.: Characteristics and sources of atmospheric mercury speciation in a coastal city, Xiamen, China, Chemosphere, 119, 530-539, https://doi.org/10.1016/j.chemosphere.2014.07.024, 2015.

---

## Author Comment (AC3)

Shi et al., reported air Hg measurements in Jan and Jul of 2012-2020, and tried to quantify the potential sources for these measured data annual trends. Generally, these long-term data are beneficial to understand the Hg emissions and air Hg variations in China, since a series of air cleaning actions taken in recent years by the government the subject is of interest; the methodology is robust; the results and discussion are presented well in the most sections. Several issues need the authors further to identify:

Response:We appreciate Prof. Feng for providing valuable comments. We have studied these comments carefully and revised the manuscript accordingly.

Section 2.2

From the authors' description, just using the Tekran 2537 without an annular denuder, the data should be mixed the signals of GOM and some PBM which with the small particle size. I suggest the authors should state clearly about their measurements. If the ratio of GOM and PBM to TGM are <5%, the authors can use the GEM to represent the TGM.

Response: We added the details of atmospheric Hg measurement in the revised manuscript **(Line 87-92)**. The Tekran 2537B/1135/1130 ambient Hg speciation system was simultaneously running in this study. During the 1 h sampling period, the GOM and PBM were first collected onto the KCl-coated quartz annular denuder and the quartz filter respectively, and GEM concentrations were measured by a Tekran 2537B Hg vapor analyzer every 5 min at a sampling flow rate of 1 L min$^{-1}$. The detection limit of GEM measurement was 0.06 ng m$^{-3}$. In the following 1 h desorbing phase, the PBM and GOM were sequentially desorbed and then quantified by the Tekran 2537B. For most of the data (70%) during the study period, the proportion of GOM and PBM in TGM was less than 5%. Thus, GEM concentrations in this study were directly compared to TGM in the following discussion.

Section 2.5

This section is very important in the whole methodology section, but the authors' description was not very clearly. Several issues need the author further confirm: one is 24h-Latitude and 24h-Longitude? What's the detail representation of these terms, the back-trajectory endpoint location during last 24 h? Another one is the air transmission. I would like to say it is the air transportation.

Response: Yes, the 24h-Latitude and 24h-Longitude interaction represents the backward trajectory endpoint location during last 24 h. We replaced the expression "air transmission" with "air transportation" in the revised manuscript. In addition, we quite agree with the importance of this section and have added more details about variable selection and factor interpretation in the manuscript **(Section 2.5 Model development).**

Section 3.1.1

Line 220-230 Given the authors only measured the GEM concentrations in Jan and Jul in each year, the authors should compare their data with the references mentioned the same month data, not the annual average data.

Response: According to our previous study, the GEM concentrations in Xiamen showed a significant seasonal variation and, January and July were typical months representing winter

and summer (Xu et al., 2015). In the revised manuscript, we added an explanation of seasonal representation **(Line 106-114)**, and compared the Jan. and Jul. data with corresponding winter and summer data from the references rather than annual average data (**Line 210-216**).

Section 3.2
In this section, the authors mainly attributed their observed Hg seasonal and diurnal cycling to the local anthropogenic emissions and long-range transport. Recently, several studies from the China cities also showed that the regional surface Hg emissions from soils and city bare regions, and Hg chemical transformations in the air of cities, and regional Hg natural surface emissions, such as from the soils and nearby the oceans. I suggest the authors to further incorporate these potential reasons in their discussion.

Response: Thanks for your advice, we have taken those potential reasons into account in our discussion. We considered the effect of Hg surface emissions especially from the ocean in summer in the discussion of regional emissions (**Line 366-371**), and considered the effect of the photo-oxidation of GEM to the diurnal variation of GEM (**Line 289-291**). What's more, we gave a more detailed explanation of the representative variables in **Section 3.3.2**. The meteorological factor represented by T and RH likely encompassed the effect of natural surface emissions, and liquid phase reduction process of reactive Hg in the atmosphere.

Section 3.3.2

This section is the key discussion parts of the whole manuscript. From the GAMs modeling results, the authors stated that the meteorological factors are the most important factor to shape the GEM variations. However, the authors mainly stated these factors' contribution which derived from the modeling. From my view, the meteorological factors influencing GEM variations by several pathways. One is that the meteorological factors drive the Hg chemical transformation, such as UV, RH are highly related to the photo-reduction and GOM formations in the air, especially in the haze, these meteorological factors playing a dominant role in GEM transformation to GOM and PBM in the air. These kinds of studies have been reported in Hefei, shanghai and Beijing. Another important role is that meteorological factors are highly related to the Hg emissions from the natural surfaces. From the current modeling results, the Hg natural emissions from the natural surfaces (e.g., soils, water, etc.) are comparable to the anthropogenic Hg emissions in China mainland. Substantial flux measurements have clearly showed that the elevated temperature and solar radiation can significantly promote the Hg re-emissions from these nature surfaces. Overall, I suggest the authors explain the cause of the contribution of meteorological factors in more detail, specifically related to the Hg emission inventory and Hg transformation mechanisms in the air, by some typical case periods of data (several tens of hour Hg, meteorological factors data) to show their interactions, not just a data presentation.

Response: These suggestions are of great help to improve the quality of the manuscript. We have made the following major revisions:

1, In the revised version, we used the min-maximum method to standardize the parameters before the model running. Then the meteorological parameters were screened mainly based on statistic judgement, and finally T, RH, and BLH were retained in the model. To better illustrate how the meteorological parameters affect GEM concentrations, we plotted the corresponding

curves of GEM with the meteorological parameters (**Figure 8, Line 420-430**). Generally, GEM concentrations increased with temperature, which was likely attributed to natural surface emissions. GEM concentrations increased with the raise of RH, possibly because the high RH environment was favorable for the liquid phase reduction of reactive Hg to GEM (Horowitz et al., 2017; Saiz-Lopez et al., 2018; Huang et al., 2019).

2. As you suggested, we have added a typical case in January 2013 to further discuss the influence of meteorological parameters on GEM (**Figure S8, Line 430-434**). In this case, we found that the daily GEM concentrations increased with the increase of T and RH as well as the decrease of BLH. The case also implies that increased temperature likely accelerated natural surface emissions and high RH was favorable for the liquid phase reduction of reactive Hg to GEM, and consequently contributed to GEM in the atmosphere.

**References**

Xu, L., Chen, J., Yang, L., Niu, Z., Tong, L., Yin, L., and Chen, Y.: Characteristics and sources of atmospheric mercury speciation in a coastal city, Xiamen, China, Chemosphere, 119, 530-539, https://doi.org/10.1016/j.chemosphere.2014.07.024, 2015.